# Earable Ω (OMEGA): A Novel Clenching Interface Using Ear Canal Sensing for Human Metacarpophalangeal Joint Control by Functional Electrical Stimulation

**DOI:** 10.3390/s22197412

**Published:** 2022-09-29

**Authors:** Kazuhiro Matsui, Yuya Suzuki, Keita Atsuumi, Miwa Nagai, Shotaro Ohno, Hiroaki Hirai, Atsushi Nishikawa, Kazuhiro Taniguchi

**Affiliations:** 1Graduate School of Engineering Science, Osaka University, Toyonaka 560-8531, Osaka, Japan; 2Graduate School of Information Sciences, Hiroshima City University, Hiroshima 731-3194, Hiroshima, Japan; 3Faculty of Human Ecology, Yasuda Women’s University, Hiroshima 731-0153, Hiroshima, Japan

**Keywords:** functional electrical stimulation, ear canal sensor, clenching, spinal cord injury, finger

## Abstract

(1) Background: A mouth-free interface is required for functional electrical stimulation (FES) in people with spinal cord injuries. We developed a novel system for clenching the human metacarpophalangeal (MP) joint using an earphone-type ear canal movement sensor. Experiments to control joint angle and joint stiffness were performed using the developed system. (2) Methods: The proposed FES used an equilibrium point control signal and stiffness control signal: electrical agonist–antagonist ratio and electrical agonist–antagonist sum. An angle sensor was used to acquire the joint angle, and system identification was utilized to measure joint stiffness using the external force of a robot arm. Each experiment included six and five subjects, respectively. (3) Results: While the joint angle could be controlled well by clenching with some hysteresis and delay in three subjects, it could not be controlled relatively well after hyperextension in the other subjects, which revealed a calibration problem and a change in the characteristics of the human MP joint caused by hyperextension. The joint stiffness increased with the clenching amplitude in five subjects. In addition, the results indicated that viscosity can be controlled. (4) Conclusions: The developed system can control joint angle and stiffness. In future research, we will develop a method to show that this system can control the equilibrium point and stiffness simultaneously.

## 1. Introduction

Functional electrical stimulation (FES) is a necessary support method for people with spinal cord injury (SCI) in their activities of daily living (ADL). Patients with an SCI at a level above the first thoracic vertebrae (Th1) often have partial or complete finger movement paralysis [1,2]. FES can be used as a means of functional compensation when the peripheral neuromusculoskeletal system does not lose function. Various FES studies have been conducted. They were applied to various human body movements: gait, cycling, or lower limb movements [3,4,5,6], reaching or upper limb movements [7,8], and grasp or hand movements [9,10,11,12,13,14,15].

However, these studies failed to mention how multiple muscles should be stimulated in a coordinated manner to achieve movement. Therefore, we have developed a novel method to improve these studies [16,17,18,19,20]. Our method constrains agonist-–antagonist muscles to control them through equilibrium point control signal and stiffness control signal: electrical agonist–-antagonist ratio (EAA ratio) and electrical agonist–antagonist sum (EAA sum). We could easily identify the neuromusculoskeletal system at various stiffness by a simple traditional system identification method using these signals. Thus, this method can help acquire an FES controller personalized to individual patients. In recent studies, we have applied our method to finger movements [21,22,23,24].

Furthermore, FES to support ADL for people with SCI requires some interfaces, such as a brain–-computer interface (BCI) [25,26,27]. In previous studies, interfaces using the mouth, such as mouthstick, respiration, and vocal commands without intrusiveness electroencephalography (EEG) [28,29,30], have been applied to patients with high-level SCIs, above the fourth cervical vertebra (C4). In recent years, some clenching interfaces for wheelchairs or computers have been reported, not just for FES [31,32,33,34].

Therefore, we propose an earphone-type clenching sensor interface for FES, unlike existing ones that require the placement of sensors around the mouth. We have developed an earphone-type ear canal movement sensor [35,36,37,38,39,40,41] and showed its application in measuring clenching occlusal force [42,43,44]. We use this earphone-type ear canal movement sensor, an earphone-type occlusal measurement device (earable Ω (OMEGA)), for an FES interface to make the mouth free from sensors by measuring the clenching motion in the ear canal. This expands the degree of freedom (DOF) referred to in a previous study [27].

Our goal in this study is to consider the effectiveness of earable Ω as an FES interface via the human middle finger metacarpophalangeal joint (MP joint) controlled by FES. This FES approach is based on our previous research. This consideration is the first phase in implementing “an earable Ω-controlled FES system for handgrip.” The following steps are taken in this study:(1)Trying to control MP joint angle by EAA ratio using earable Ω;(2)Trying to control MP joint stiffness by EAA sum using earable Ω.

(1) and (2) aim to reveal that the proposed system can control joint angle and stiffness.

## 2. Materials and Methods

### 2.1. Earable Ω

“Earable” is a coined term that combines “wearable” with “ear”. “Ω (OMEGA)” is an abbreviation of “Occlusal MEasurement for Generating Articular movements”. Furthermore, Ω reminds us of the shape of the mouth when we clench, and it is also the ohm symbol in the SI unit of electrical resistance, which can remind us of electrical stimulation. Earable Ω is an earphone-type wearable device that has a reflective photosensor in the earpiece’s center. It can measure clenching force by sensing ear canal deformation [42,43,44]. Earable Ω is shown in Figure 1. Some clenching sensors, such as the T-scan (Nitta Co., Osaka, Japan) [45] and occlusal force gauge meter GM10 (NAGANO KEIKI Co., Ltd., Nagano, Japan) [46], have limited applications because they must be inserted into the mouth and need some external hold settings. However, earable Ω overcomes these issues.

Clenching is performed on the temporal and masticatory muscles, including the masseter muscle and the temporomandibular joint. Clenching causes changes in the shape of the ear canal near the masticatory muscles and temporomandibular joint. Earable Ω optically and noninvasively measures changes in ear canal shape during clenching. A small QRE1113 photosensor (Fairchild Semiconductor International Inc., Sunnyvale, CA, USA) is used in earable Ω to determine the change in the ear canal shape. This photosensor incorporates a light-emitting diode (LED) with an emission wavelength of 940 nm and a phototransistor. The sensor irradiates infrared light onto the ear canal’s skin, and the phototransistor receives the reflected light to measure changes in the ear canal’s shape. The sensor’s output increases when the amount of reflected light from the ear canal increases. Similarly, the output decreases when the amount of reflected light decreases. The output offset voltage of the ear sensor can be adjusted using a variable resistor, and the LED is equipped with a pulse wave generator to control its emission [42].

### 2.2. Experiment 1: Controlling Joint Angle Using Earable Ω as an Interface

In this experiment, the EAA ratio was changed by the earable Ω signal to confirm that the earable Ω signal could be used to control the MP joint angle.

#### 2.2.1. EAA Ratio and EAA Sum for Joint Angle Control

We have proposed control signals that focus on the coordination between the agonist and antagonist muscles: the EAA ratio and the EAA sum. These signals were inspired by the agonist-–antagonist muscle ratio (AA ratio) and the agonist-–antagonist muscle sum (AA sum) and contributed to the joint equilibrium point and joint stiffness [47,48]. The AA ratio and AA sum are given by Equations (Equation 1) and (Equation 2). The flexor and extensor muscle activities of the AA muscles obtained from electromyography (EMG) are defined as $m_{e}$ and $m_{f}$, respectively. Notably, *r* and *s* were only used to indicate that $r_{E}$ and $s_{E}$ refer to them in this study.
(1)$$r=\frac{m_{e}}{m_{e}+m_{f}}$$
(2)$$s=m_{e}+m_{f}$$

The proposed signals, the EAA ratio and EAA sum, are given by Equations (Equation 3) and (Equation 4). The normalized FES intensity to the flexor and extensor are defined as $V_{e}$ and $V_{f}$, respectively. To minimize the differences in the characteristics of the flexor and extensor, the stimulus voltage values are normalized. The subjects sat on a chair with their elbow joint kept at 135${}^{{}^{\circ}}$ and their shoulder joint kept at 45${}^{{}^{\circ}}$, and the finger end was connected to a fixed force gauge ZP-50 N (IMADA Co., Ltd., Aichi, Japan) at an MP joint angle of 180${}^{{}^{\circ}}$. The maximum stimulus voltages $V_{\mathrm{emax}}^{\prime }$ and voltage $V_{\mathrm{fmax}}^{\prime }$ were determined as the voltages just before the subjects felt pain. The minimum stimulus voltages $V_{\mathrm{emin}}^{\prime }$ and voltage $V_{\mathrm{fmin}}^{\prime }$ were determined at the point where muscle contraction commenced, and the threshold was 0.05 kgf. These were used for normalization (Equations (Equation 5) and (Equation 6)), where $V_{e}^{\prime }$ and $V_{f}^{\prime }$ are the voltages prior to normalization. For details, refer to previous studies [16,17,19].
(3)$$r_{E}=\frac{V_{e}}{V_{e}+V_{f}}$$
(4)$$s_{E}=V_{e}+V_{f}$$
(5)$$V_{f}=\frac{V_{f}^{\prime }-V_{\mathrm{fmin}}^{\prime }}{V_{\mathrm{fmax}}^{\prime }-V_{\mathrm{fmin}}^{\prime }}$$
(6)$$V_{e}=\frac{V_{e}^{\prime }-V_{\mathrm{emin}}^{\prime }}{V_{\mathrm{emax}}^{\prime }-V_{\mathrm{emin}}^{\prime }}$$

#### 2.2.2. The Developed System for Experiment 1

The developed system can measure clenching activity *C* ( 0.0≤C≤1.0), which varies linearly with clenching force by earable Ω [42]. The clenching force is affected by the occlusion status of the upper and lower teeth. The analog data are sent to a microcomputer ARDUINO©Mega 2560 (Arduino) for signal processing and to data logger PowerLab (ADInstruments Co., Colorado Springs, CO, USA) for display on the PC monitor. The microcomputer is set to C=1 when the subjects achieve maximum clenching and set to C=0 when they make their upper and lower teeth to be barely touched. No clenching is defined as a situation in which C=0, or when subjects make their upper and lower teeth barely touch. In this system, the EAA ratio $r_{E}$ is controlled by clenching activity *C* with the EAA sum $s_{E}$ being constant at 0.5. The stimulation signals are processed by multiplying a carrier sine wave with frequency f= 1000 Hz and modulated at *C*. $r_{E}$ was *C* in this experiment. These signals are given by Equations (Equation 7) and (Equation 8). This signal processing image is shown in Figure 2. These signals are transmitted to an electro-stimulator DPS-5508 (Physio-tech Co., Ltd., Tokyo, Japan) via an electronic circuit to stimulate the muscles.
(7)$$V_{f}^{\prime }\left(t\right)=\left\{V_{\mathrm{fmin}}^{\prime }+\left(V_{\mathrm{fmax}}^{\prime }-V_{\mathrm{fmin}}^{\prime }\right)\cdot 1/2\right\}\cdot \left\{1-C\left(t\right)\right\}\cdot \sin\left(2\pi ft\right)$$
(8)$$V_{e}^{\prime }\left(t\right)=\left\{V_{\mathrm{emin}}^{\prime }+\left(V_{\mathrm{emax}}^{\prime }-V_{\mathrm{emin}}^{\prime }\right)\cdot 1/2\right\}\cdot C\left(t\right)\cdot \sin\left(2\pi ft\right)$$

#### 2.2.3. Experimental Environment and Method

The experimental environment is shown in Figure 3. The target was the horizontal plane human left middle finger MP joint movement. The wrist joint, proximal interphalangeal (PIP) joint, and distal interphalangeal (DIP) joint were fixed. The subjects sat on a chair with their elbow joint kept 135${}^{{}^{\circ}}$ and shoulder joint kept 45${}^{{}^{\circ}}$ They faced towards the front and did not watch their hands to eliminate visual feedback and only watched the *C* value on the PC monitor. The initial MP joint was at a natural angle for the subject. The target muscles were the flexor digitorum superficialis and extensor digitorum. The earable Ω was worn on the right ear. The MP joint angle was acquired using a goniometer (Biometrics Ltd., Ladysmith, VA, USA) since the extension direction was positive, and this analog data was sampled using PowerLab with clenching activity data at a rate of 1000 Hz.

Six healthy adult Subjects A–F volunteered to participate in the experiment. Subjects were first instructed to maintain “no clenching” before the stimulation with $r_{E}=0$. Second, data sampling began once the experimenter confirmed that a plateau of MP joint flexion by stimulation with no clenching had been reached. Finally, subjects were only asked to clench “freely” for more than 10 s. Notably, the subjects’ alphabets were not in any particular order, with subject E being the last one. As a result, the protocol was revised in light of the results of other subjects in the experiment with Subject E. Because Subjects A–D and F were clenched to C=1, only Subject E performed an additional trial in addition to the previous trial. During the trial, he was told not to clench more than C=0.5. A normalized cross-correlation function was used in the analysis to compare the similarity between C as input and MP joint angles as output. The degree of similarity was calculated using MATLAB 2017b (MathWorks Inc., Natick, MA, USA) by normalizing the zero-lag autocorrelation to 1 and taking the maximum value of the calculated values. The outcomes for each subject were individually and thoroughly discussed.

### 2.3. Experiment 2: Controlling Joint Stiffness Using Earable Ω as an Interface

In this experiment, the EAA sum was changed by an earable Ω signal to confirm that the earable Ω signal could be used to control MP joint stiffness.

#### 2.3.1. The Developed System for Experiment 2

In this experiment, the system used was the same as in Section 2.2.2, except that the EAA sum $s_{E}$ was controlled by clenching activity *C* with the EAA ratio $r_{E}$ being constant at 0.5. From this, Equation (Equation 7) is changed to Equation (Equation 9).
(9)$$V_{f}^{\prime }\left(t\right)=\left\{V_{\mathrm{fmin}}^{\prime }+\left(V_{\mathrm{fmax}}^{\prime }-V_{\mathrm{fmin}}^{\prime }\right)\cdot 1/2\right\}\cdot C\left(t\right)\cdot \sin\left(2\pi ft\right)$$

#### 2.3.2. Acquirement of Joint Stiffness

The joint stiffness was acquired as described in [17,20]. In this method, the chirp wave-generated force $f_{e}$ at constant $s_{E}$ was inputted and the joint angle was measured as the output. Notably, the chirp wave was modulated exponentially from 0.1 Hz to 10 Hz for 30 s. Thus, a transfer function was identified from the input-–output relationship. In this method, the joint movement system was assumed to be a second-order system, as in Equation (Equation 10), where $K_{p}$ is a proportional gain, $\omega_{n}$ is the natural angular frequency, and ζ is the damping ratio. The transfer function was identified from the input-–output relationship using the System Identification Toolbox in MATLAB 2017b. This toolbox calculates normalized root mean square error (NRMSE)(%) for evaluation of the goodness of fit between the output and estimated values by transfer functions and the reliability of transfer functions. Although it has no general threshold, it can be used for relative comparisons. For simplicity, we assumed that the fingers were bars of uniform density. We considered the change in the MP joint angle $\theta^{\prime }$ when an external force $f_{e}$ was applied perpendicular to the finger (Figure 4). Assuming that the finger length is *l*, the moment of inertia is *I*, the viscosity is *D*, and the stiffness is *K*, then the equation of motion for rotation is Equation (Equation 11) and its transfer functions are Equations (Equation 12). From Equations (Equation 10) and (Equation 12), $K_{p}$, $\omega_{n}$, and ζ are represented by Equations (Equation 13)–(Equation 15), respectively. These indicate that $K_{p}$ and $\omega_{n}$ are indexes of stiffness.
(10)$$G\left(s\right)=K_{p}\cdot \frac{\omega_{n}^{2}}{s^{2}+2\zeta \omega_{n}s+\omega_{n}^{2}}$$
(11)$$I\ddot{\theta^{\prime }}+D\dot{\theta^{\prime }}+K\theta^{\prime }=f_{e}l$$
(12)$$G\left(s\right)=\frac{l}{Is^{2}+Ds+K}$$
(13)$$K_{p}=\frac{l}{K}$$
(14)$$\omega_{n}=\sqrt{\frac{K}{I}}$$
(15)$$\zeta =\frac{D}{2\sqrt{IK}}$$

Furthermore, $\zeta \omega_{n}$ is represented by Equation (Equation 16) and is assumed as the viscosity index.
(16)$$\zeta \omega_{n}=\frac{D}{2I}$$

#### 2.3.3. Experimental Environment and Method

The experimental environment and experimental scene are shown in Figure 5 and Figure 6. As in Experiment 1, the target was the horizontal plane human left middle finger MP joint movement; wrist, PIP, and DIP joints were fixed. The subjects sat on a chair with their elbow joint kept 135${}^{{}^{\circ}}$. They faced towards the front and did not watch their hands to eliminate visual feedback. The initial MP joint was at a natural angle for the subjects. The target muscles were the flexor digitorum, superficialis, and extensor digitorum. The earable Ω was worn on the right ear. Phantom Premium (3D Systems Inc., Rock Hill, SC, USA) was used to apply $f_{e}\left(t\right)$ as a robot arm. Phantom Premium was attached to the end point of the finger through a force sensor USL06-H5-200N (Tec Gihan Co., Ltd., Kyoto, Japan). In this experiment, the extension direction was a positive force. The MP joint angle was acquired using a goniometer, and this analog data was sampled by PowerLab with clenching activity data and force data at a rate of 1000 Hz. Five healthy adult Subjects A–E volunteered to participate in the experiment. They were the same subjects as in Experiment 1, except for Subject F, because Subject F’s data could not be obtained accurately due to some system failures. They were instructed to maintain constant clenching activity, confirming the PC1 monitor. Three patterns of clenching activity were used. $s_{E}$ was *C* in this experiment. Before 2 s of applying extra force by Phantom Premium, clenching and stimulation were started.

First, the mean and standard deviation (SD) between subjects was calculated as statistical values to confirm that the indexes of stiffness and viscosity tended to change in response to clenching. For statistical evaluation, multiple comparisons were performed using the Bonferroni test (significance level is 0.05). HAD [49] was the software used. Furthermore, because statistical analysis may not be enough, within-subject changes were evaluated individually and thoroughly.

#### 2.3.4. An Example of Maintaining *C*

To confirm the degree of difficulty in maintaining *C*, the time series *C* value was recorded in Subject E as the final subject. An example of maintaining *C* of Subject E is shown in Figure 7. The time scale was normalized. When the target of *C* was 0.0, the variation in *C* was large but did not cover the target value of C=0.5. This is because it was the subject’s first trial and the subject had not been familiarized with it. In addition, no clenching position was an unnatural posture for the subject and gradually closing and opening the mouth were repeated due to the subject’s fatigue. When the target of *C* was 0.5, *C* was almost flat because the posture with C=0.5 was natural for the subject. When the target of *C* was 1.0, *C* was also almost flat because *C* was clipped at 1.0. Although the natural posture of each subject varies, this experimental environment can be assumed to produce three patterns of clenching states: low, middle, and high. In this experiment, the averaged stiffnesses of the whole trial in each state were obtained.

#### 2.3.5. Information on Subjects

The information on subjects is shown in Table 1. The average age was 22.3 years and SD was 0.75 years.

## 3. Results

### 3.1. Experiment 1

Table 2 displays the maximum values of the normalized cross-correlation function, and Figure 8 displays all results. All subjects except Subject E performed only a first-time trial, and only Subject E had two trials. The time scales were normalized.

The maximum values of the normalized cross-correlation function are all greater than 0.76. Subject A clenched and increased *C* from about 0.0 to 1.0 all at once and maintained it, which later reduced to about 0.25. His MP joint angle followed this trajectory with delay and hysteresis in the range of approximately 150${}^{{}^{\circ}}$ to 180${}^{{}^{\circ}}$. Subject B clenched and increased *C* from about 0.0 to 1.0 in stages and maintained it, which later reduced to about 0.20 with an undershoot. Except for steep changes in trajectory, his MP joint angle followed a path in the range of approximately 130${}^{{}^{\circ}}$ to 170${}^{{}^{\circ}}$.

Subject C clenched and increased *C* from about 0.1 to 1.0 all at once and maintained it, including a short reduction period, which later reduced to about 0.0 and then gradually increased to 0.2. His MP joint angle followed this trajectory in the range of approximately 130${}^{{}^{\circ}}$ to 205${}^{{}^{\circ}}$ until a short clenching reduction period. However, after this period, the MP joint angle was reduced to about 190${}^{{}^{\circ}}$ even though *C* was maintained at 1.0. Furthermore, *C* was reduced from 1.0 to 0.0 all at once, the MP joint angle was decreased gradually with an initial rapid small reduction, and finally reduced to about 120${}^{{}^{\circ}}$, which was smaller than the initial MP joint angle. Subject D clenched and increased *C* from about 0.0 to 1.0 relatively slowly with initially small vibrations and maintained it, which later reduced to about 0.30. His MP joint angle followed this trajectory with the initial *C* vibrations damped until a clenching reduction in the range of approximately 155${}^{{}^{\circ}}$ to 190${}^{{}^{\circ}}$. Some small vibrations more than 180${}^{{}^{\circ}}$ were observed in the extension term. Subsequently, *C* was reduced from about 1.0 to 0.3 all at once, and the MP joint angle was almost not reduced after the initial rapid small reduction. On Subject E’s the first time, Subject E clenched and increased *C* from about 0.0 to 1.0 relatively slowly with initial small vibration and maintained it, which later reduced to about 0.0 in stages, and then increased to about 0.3 slowly. His MP joint angle followed this trajectory with *C* vibrations and steep changes damped in the range of approximately 125${}^{{}^{\circ}}$ to 170${}^{{}^{\circ}}$. On Subject E’s second time, Subject E clenched and increased *C* from about 0.0 to 0.5 slowly and maintained it with small vibration, which later reduced to about 0.0. His MP joint angle followed this trajectory with *C* vibrations and steep changes damped, but the change in angle was small. Subject F clenched and increased *C* from about 0.1 to 1.0 all at once and maintained it, which later reduced to about 0.0 with a steep change. His MP joint angle followed this trajectory until a clenching reduction in the range of approximately 125${}^{{}^{\circ}}$ to 195${}^{{}^{\circ}}$. However, after it, the MP joint angle was reduced slowly with an initial rapid small reduction, even though *C* was reduced from 1.0 to 0.0 all at once. The MP joint angle was finally reduced to about 110${}^{{}^{\circ}}$. which was smaller than the initial MP joint angle.

### 3.2. Experiment 2

Figure 9 and Table 3 show the results of the transfer function parameters, and Figure 10 shows the results of NRMSE. For the results of average, $\omega_{n}$ and $\zeta \omega_{n}$ were increased with the target of *C* while $K_{p}$ was decreased with the target of *C*. Only the target of *C* is 0 and 1 was significant for the others, and $K_{p}$ was not significant.

For the results of Subjects A and B, $\omega_{n}$ was increased and $K_{p}$ was decreased within the target of *C*. For Subject C’s results, $\omega_{n}$ slightly increased, to the extent where it could be said to be equivalent with *C*’s target range of 0.0–0.5. This parameter was increased by a small quantity with a *C* target range of 0.5–1.0. However, $K_{p}$ was decreased slightly, to the extent that it could be said to be equivalent to the *C* target range of 0.0–0.5. This parameter also decreased by a small quantity with the *C* target range of 0.5–1.0. For Subject D’s results, $\omega_{n}$ was increased for the *C* target range of 0.0–0.5. For the *C* target range of 0.5–1.0, $\omega_{n}$ was increased by a small quantity. In contrast, $K_{p}$ was decreased for the *C* target range of 0.0–0.5. For a *C* target range of 0.5–1.0, $K_{p}$ was increased by a small quantity and it was almost equivalent to the target of *C*. For Subject E’s results, $\omega_{n}$ increased slightly with the target of *C*, to the extent that it could be said to be equivalent to the *C* target range of 0.0–0.5. For a *C* target range of 0.5–1.0, $\omega_{n}$ increased. However, $K_{p}$ decreased by a large quantity for a *C* target range of 0.0–0.5 and it decreased by a smaller quantity for the *C* target range of 0.5–1.0.

Furthermore, the results of $\zeta \omega_{n}$ on Subjects A, B, and D increased with *C*’s target. The results of $\zeta \omega_{n}$ on Subject C increased with the *C* target by a small quantity. The results of $\zeta \omega_{n}$ on Subject E increased with a *C* target range of 0.0–0.5. For a *C* target range of 0.5–1.0, $\zeta \omega_{n}$ decreased with *C* target value. However, $\zeta \omega_{n}$ for a *C* target value of 1.0 was greater than that of 0.0.

From the NRMSE results, the average of all results was 58.2% with a standard deviation of 12.9% and the value obtained by subtracting the standard deviation from the mean was 45.3%. The result of Subject D with a *C* target of 0.0 and those of Subject E for *C* targets of 0.5 and 1.0 were lower than this difference.

## 4. Discussion

### 4.1. Experiment 1

While the perfect match is 1, all of the maximum values of the normalized cross-correlation function are greater than 0.76. As a result, *C* can control the joint angle. However, because there is no universal standard for cross-correlation, the results for each subject are discussed separately and in-depth.

All results showed that the initial angles were not always the same for each subject, which was caused by the normalization of the stimulus voltage values. The flexor was almost exclusively at nearly C=0.0, and each subject’s maximum voltage determinations of the flexor were based on their subjective pain perception rather than force or motion. They did not necessarily realize the same motion as a result.

From Subject A’s results, the joint angle can be controlled with some delay and hysteresis. The delay is considered to be caused by the delay of the device and the delay of the human neuromusculoskeletal system. The system is considered to be a cascade coupling of second-order delay and wasted time, which can cause delays [16,17]. The hysteresis is probably caused by the normalizing method of the stimulus voltage values. The muscle forces at the maximum voltage are not balanced: in some non-narrow areas of *C*, single-muscle contraction seems to have occurred. Gradual angle control is also possible when considering Subject B’s results, which are probably caused by the characteristics of the human neuromusculoskeletal system. Because the second-order delay system performed as a low-pass filter, steep changes were damped, indicating that the device is not susceptible to noise. Subject C’s results showed that steep increase in *C* caused hyperextension of the MP joint. This hyperextension produces tension in the accessory collateral ligaments, which is considered to increase joint stability and balance the tension in tissues in the direction of flexion [50,51]. However, the short clenching reduction period caused reduced stability. The acceleration after this period is insufficient to return to the hyperextension angle. From these, the maintained angle is considered to be reduced to about 190${}^{{}^{\circ}}$ from 205${}^{{}^{\circ}}$. From the slow reduction after maintaining of clenching, we consider that hyperextension causes some change in the characteristics of the neuromusculoskeletal system. For Subject D’s results, the small vibrations in the initial rising are considered to be damped by low-pass filter characteristics. The small vibrations in the MP joint angle are due to hyperextension, as they were observed in over 180${}^{{}^{\circ}}$ areas. The small reduction in the MP joint angle is probably caused by the high stability in hyperextension and insufficient reduction in *C*. From the first-time result for Subject E, the steep fluctuations such as noise were dampened, and the gradual control commands were well followed. From the second-time result for Subject E, the relationship between *C* and the MP joint angle was nonlinear because the MP joint angle range was smaller than that of the first-time result, which was caused by ambiguity in the calibration method of voltage values. We have to improve the calibration method by defining the maximum voltages in terms of force or motion or using only the *C* area with a linear characteristic of force or motion. From Subject F’s results, good tracking performance was confirmed, and as with Subject C’s results, a phenomenon that appeared to be a change in neuromusculoskeletal characteristics due to hyperextension was also observed.

### 4.2. Experiment 2

From the experimental environment, *C*’s stability was not certain, but some trends in the relationship between the target value of *C* and each parameter can be discussed because the classification was roughly divided into high, middle, and low. We believe that earable Ω can control stiffness if there is a consistent trend between *C* and each parameter.

There was a trend that $\omega_{n}$ and $\zeta \omega_{n}$ increased with the target of *C* and $K_{p}$ decreased with the target of *C*. While some results were significant, the SDs of these variables and those in previous studies [23] show that these parameters vary between individuals, and five subjects do not lend credibility to statistical analysis. As a result, we examined within-subject changes individually and thoroughly.

First, we discuss the stiffness from the results of $\omega_{n}$ and $K_{p}$. In the results of $\omega_{n}$ and $K_{p}$ on Subjects A and B, as the target of *C* increases, the parameter $\omega_{n}$, which increases with stiffness (Equation (Equation 14)), increases, and the parameter $K_{p}$, which decreases with stiffness (Equation (Equation 13)), decreases. In the results of $\omega_{n}$ and $K_{p}$ on Subject C, as the target of *C* increases, the parameter $\omega_{n}$ increases, and when $\omega_{n}$ increases by a small quantity, $K_{p}$ decreases by a small quantity in response to it. In the results of $\omega_{n}$ and $K_{p}$ in Subject D, if we ignore the result when the target of *C* is 0.5, in which NRMSE is low, then as the target of *C* increases, the parameter $\omega_{n}$ increases and $K_{p}$ decreases. In the results of $\omega_{n}$ and $K_{p}$ on Subject E, although $\omega_{n}$ increases by a small quantity, $K_{p}$ decreases by a large quantity. This could be caused by signal noise and low reliability, because NRMSEs for *C* targets of 0.5 and 1.0. are low. However, the exact cause is unknown and should be carefully examined to ensure that similar phenomena do not occur in the future. Therefore, these results indicate that clenching controls stiffness.

Next, we discuss the viscosity from the results of $\zeta \omega_{n}$. From the results of $\zeta \omega_{n}$ on Subjects A–D, although the amount of the change in Subject C is small, these parameters increased with *C*’s target value. From the results of $\zeta \omega_{n}$ on Subject E, we were unable to determine whether this parameter decreased or increased with *C*’s target value. This could also be caused by signal noise and low reliability, because NRMSEs for *C* target values of 0.5 and 1.0 are low. From these results, clenching may control viscosity.

The summary of these discussions is shown in Table 4.

### 4.3. Summary of Discussion

Based on these discussions, we believe that earable Ω may be able to control the MP joint angle and stiffness, as well as viscosity, without the use of any equipment around the mouth, whereas other studies require the use of some equipment around the mouth [31,32,33,34].

### 4.4. Future Outlook

Notably, humans control the equilibrium point and stiffness in their voluntary movements at the same time [52]. Therefore, a simple two-DOF interface [27] and an interface that can control “equilibrium point trajectory” and “strain for stiffness” simultaneously for natural smooth one-DOF movements are required. When a subject uses two synchronized earable Ωs on both ears, as already realized in [40], $C_{r}$ is defined as the right side clenching and $C_{l}$ is defined as the clenching of the left side. For example, $r_{C}$ and $s_{C}$ can be defined as Equations (Equation 17) and (Equation 18) from $C_{r}$ and $C_{l}$. Patients with SCI can simultaneously control the equilibrium point and stiffness using this interface.
(17)$$r_{C}=\frac{C_{r}}{C_{r}+C_{l}}$$
(18)$$s_{C}=C_{r}+C_{l}$$

## 5. Conclusions

We developed an FES system with a novel clenching interface using an ear canal sensor, earable Ω, for the MP joint of patients with SCI. To validate this system, we conducted separate experiments for controlling the MP joint angle and stiffness. The developed system used equilibrium point and stiffness control signals for FES: EAA ratio and EAA sum. The joint angle was acquired using an angle sensor, and the joint stiffness was acquired by system identification using external force of a robot arm. Six and five subjects participated in each experiment, respectively. Our findings showed that the proposed system can control joint angle and stiffness, as well as viscosity. Furthermore, we propose that this system can control the equilibrium point and stiffness simultaneously, which will be the subject of a future study. However, this study has some limitations: problems in voltage calibration and a lack of rigorous verification of the correspondence between clenching and stiffness or viscosity. We expect to address these issues in future studies.

## Figures and Tables

**Figure 1 sensors-22-07412-f001:**
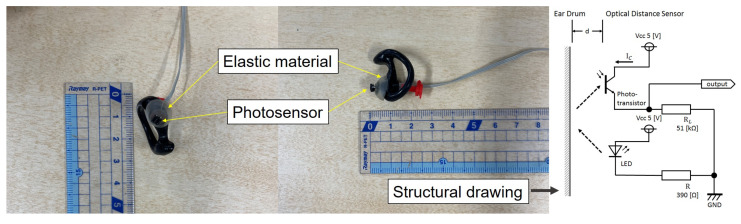
Earable Ω.

**Figure 2 sensors-22-07412-f002:**
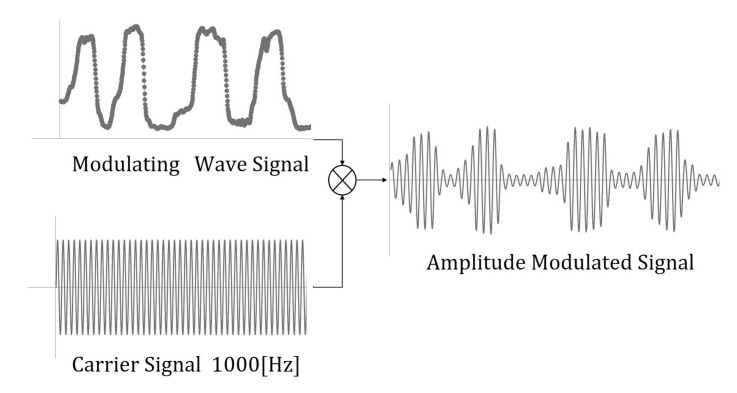
Signal processing image.

**Figure 3 sensors-22-07412-f003:**
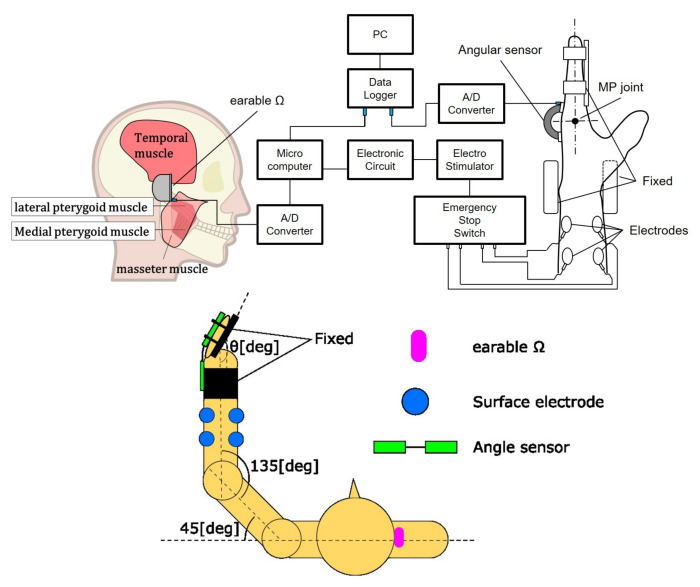
Upper figure shows the system configuration diagram. Lower figure shows an experimental setup for controlling the joint angle using earable Ω as an interface, top view. The stimulation at the surface electrode is via the emergency stop switch for the safety of subjects.

**Figure 4 sensors-22-07412-f004:**
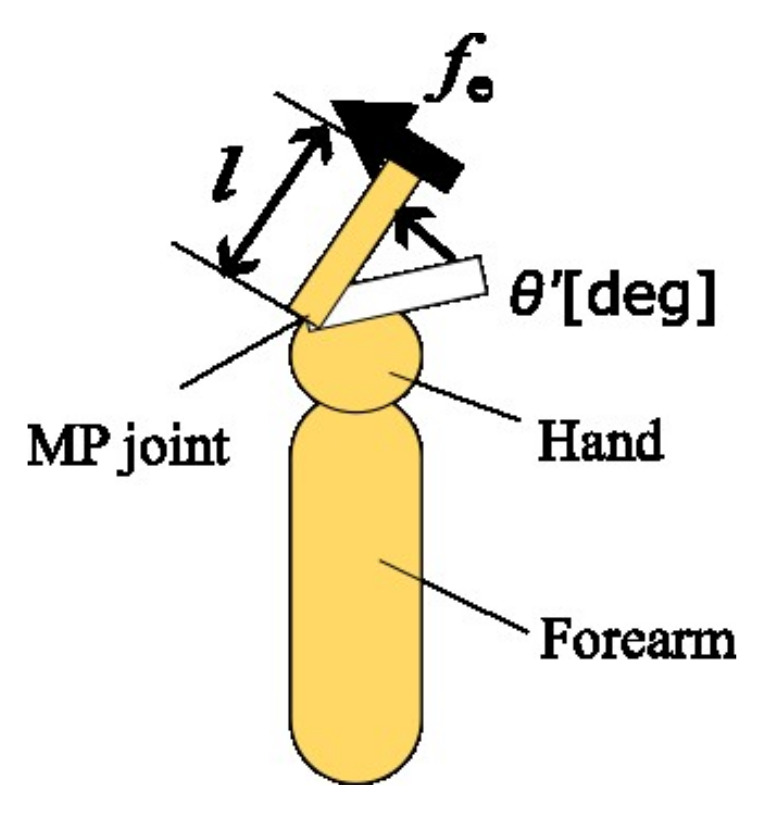
A 1-link model of MP joint.

**Figure 5 sensors-22-07412-f005:**
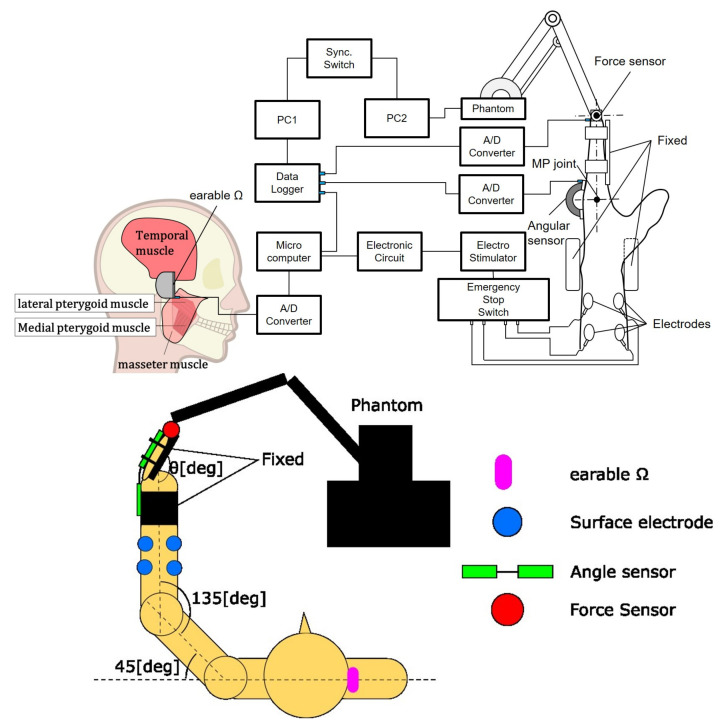
Upper figure shows the system configuration diagram. Lower figure shows an experimental setup for controlling joint stiffness using earable Ω as an interface, top view. The stimulation at the surface electrode is via the emergency stop switch for the safety of subjects.

**Figure 6 sensors-22-07412-f006:**
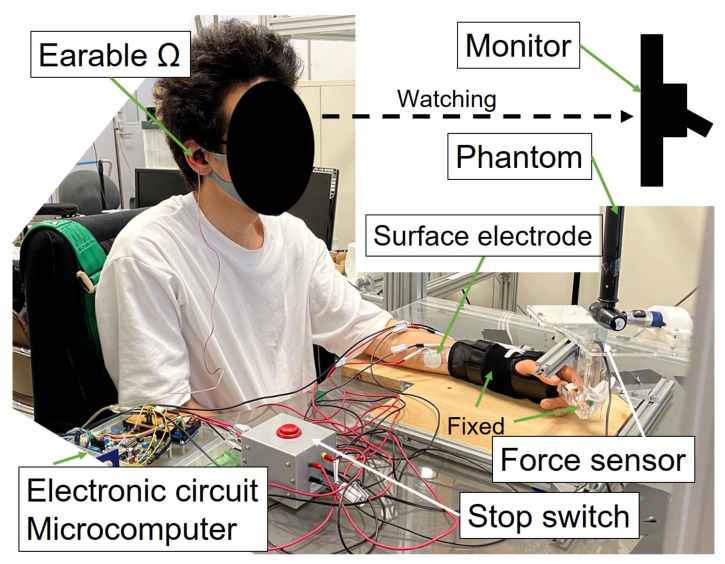
Experimental scene.

**Figure 7 sensors-22-07412-f007:**
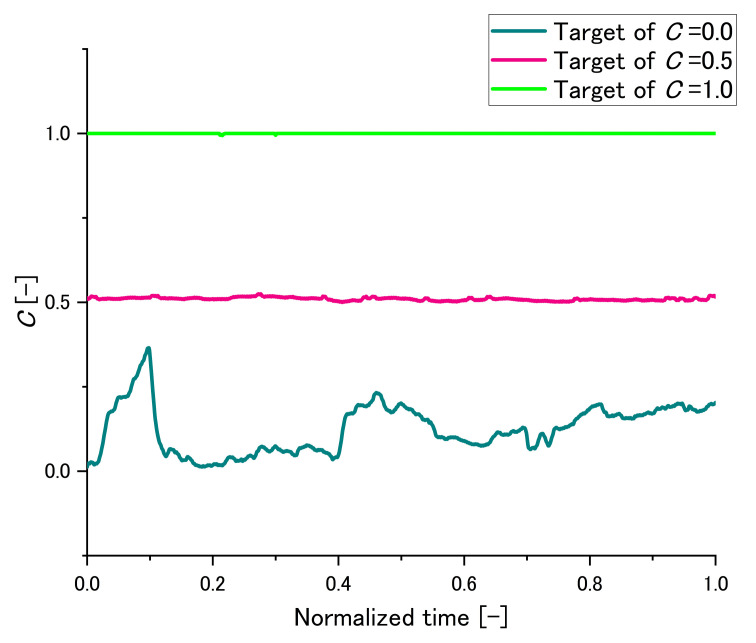
An example of maintaining *C* of Subject E.

**Figure 8 sensors-22-07412-f008:**
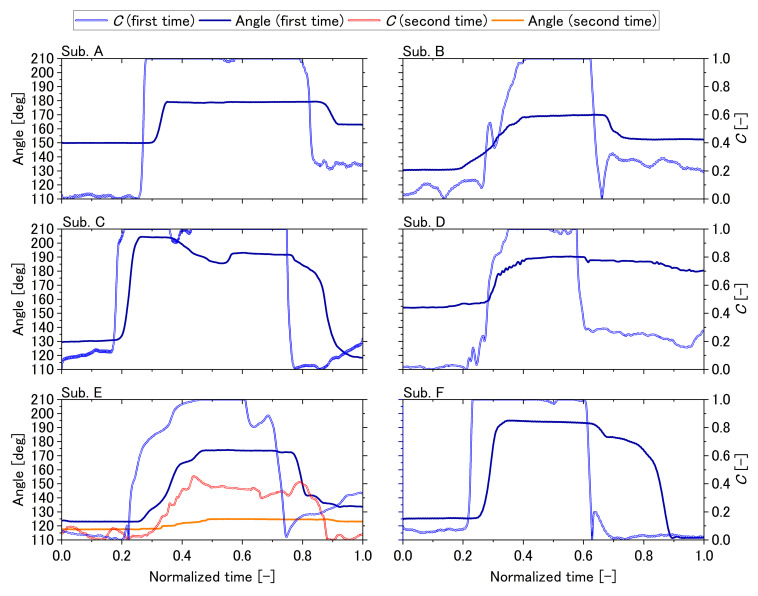
The results of the six subjects for controlling the joint angle using earable Ω as an interface.

**Figure 9 sensors-22-07412-f009:**
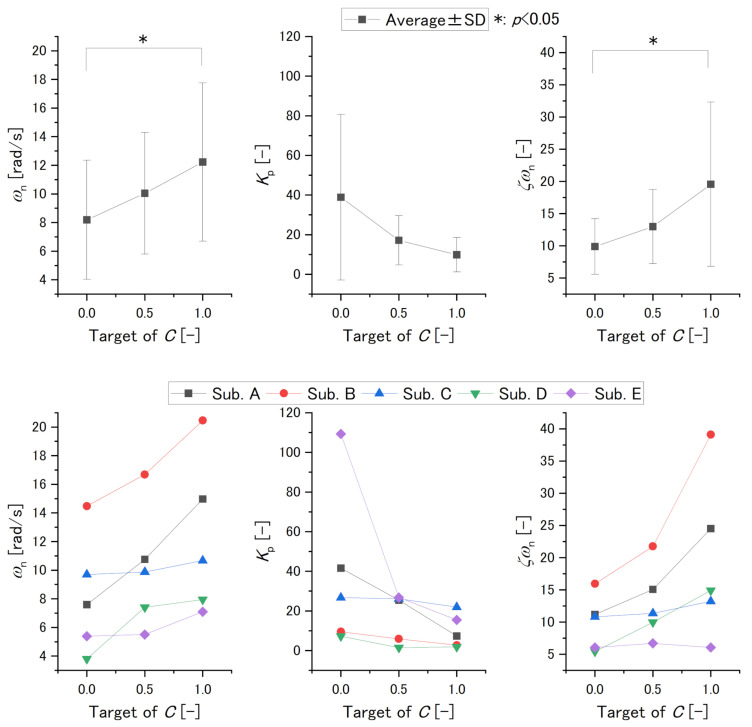
The results of $\omega_{n}$ (**left**), $K_{p}$ (**center**), $\zeta \omega_{n}$ (**right**). Upper figures are averages and SDs. Lower figures are individual values.

**Figure 10 sensors-22-07412-f010:**
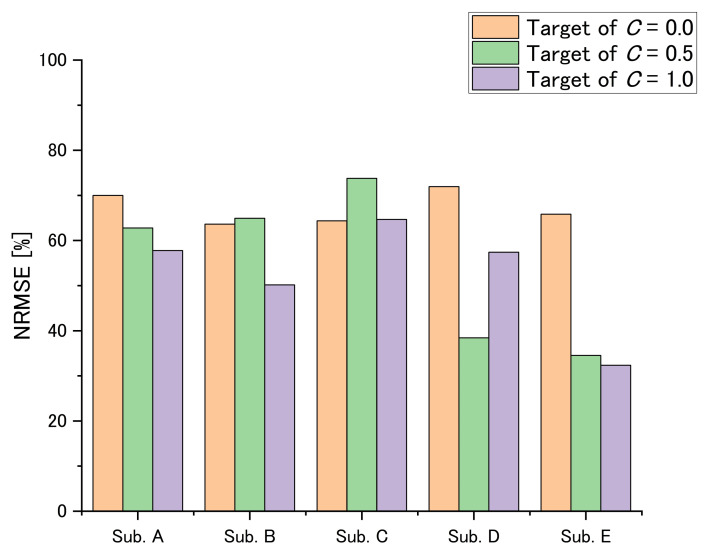
The results of NRMSE.

**Table 1 sensors-22-07412-t001:** Information on subjects.

Subjects	Age (Years)	Gender	Dominant Hand
A	22	Male	Right
B	22	Female	Right
C	22	Male	Right
D	24	Male	Right
E	22	Male	Right
F	22	Male	Right

**Table 2 sensors-22-07412-t002:** The maximum value of normalized cross-correlation function.

Subjects	The Maximum Value of Normalized Cross-Correlation Function [-]
A	0.84
B	0.81
C	0.88
D	0.76
E (first time)	0.80
E (second time)	0.84
F	0.79

**Table 3 sensors-22-07412-t003:** Statistics values.

			Averages			SDs	
	Target of C [-]	0.0	0.5	1.0	0.0	0.5	1.0
Variable							
**$\omega_{n}$** [rad/s]		8.2	10.0	12.2	4.2	4.2	5.5
**$K_{p}$** [-]		38.9	17.1	9.9	41.8	12.4	8.7
**$\zeta \omega_{n}$** [-]		9.9	13.0	20.0	4.3	5.8	12.7
			**p values** [-]				
	**Target of C** [-]	**0.0–0.5**		**0.0–1.0**		**0.5–1.0**	
**Variable**							
$\omega_{n}$		0.08		0.003		0.05	
$K_{p}$		0.16		0.06		0.60	
$\zeta \omega_{n}$		0.34		0.01		0.06	

**Table 4 sensors-22-07412-t004:** The summary of each variable.

Variable	Significant Difference	Individual Evaluation
		(Relationship of Variable and *C*’s Target )
$\omega_{n}$	*C*’s target of 0.0 and 1.0	Positive
$K_{p}$	No significant difference	Negative
$\zeta \omega_{n}$	*C*’s target of 0.0 and 1.0	Positive or cannot determine

## Data Availability

The data are not publicly available due to privacy or ethical issues.

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
