# Peer review of "Earable Ω (OMEGA): A Novel Clenching Interface Using Ear Canal Sensing for Human Metacarpophalangeal Joint Control by Functional Electrical Stimulation"

_sensors, 2022, doi:10.3390/s22197412_

Round 1

Reviewer 1 Report

The authors developed a system for clenching the human metacarpophalangeal (MP) joint using an earphone-type ear canal movement sensor. Experiments were performed to control the joint angle and joint stiffness using the developed system. Findings showed that the proposed system could control joint angle, stiffness, and viscosity. While the joint angle could be controlled well by clenching with some hysteresis and delay in three subjects, it could not be controlled relatively well after hyperextension in other subjects, which revealed the calibration problem and the change in the characteristics of the human MP joint caused by hyperextensions. The joint stiffness increased with the clenching amplitude in five subjects. This system has limitations in voltage calibration and there is a lack of rigorous verification of correspondence between clenching and stiffness or viscosity. Overall this paper is interesting to read. Here are a few minor comments that are needed to be addressed.

1.       It would be better to add the mechanism and structure of the earphone-type ear canal in Figure 1 for better understanding.

2.       Please mention each variable in this work and comparison of the results in tabular form for better understanding.

3.       Related to sections 2.3.3 “experimental environment and method” and 2.3.4 “An example of maintaining “, please add photos or videos for the visualization of the experimental setup in real-time.

Author Response

Dear Reviewer 1.

Thank you for taking the time out of your busy schedule to read our manuscript.
We sincerely appreciate your insightful comments.
We have renumbered and reorganized the comments made by each of the two reviewers into a matrix, and have included our responses to each comment.

Reviewer 2 Report

This paper proposed a novel clenching interface for human metacarpophalangeal (MP) joint control. An earphone-type sensor was developed to detect ear canal movement while clenching. And two experiments were designed to validate the performance of the sensor. The proposed system can control the MP joint angle and stiffness.

The detailed comments are as follows:

Ø  The introduction section needs to be reorganized; the article cites a large amount of literature, especially between lines 25 and 26, where the authors cite a large amount of literature that summarizes the current state of research in a simple description. The authors should have focused their discussion more on papers that are closely related to the purpose of this paper. In addition, a large number of papers published before 2000 appear in the references, but we would prefer that the authors focus more on high-quality papers published in recent years when doing their literature research.

Ø  The introduction section could be written in several paragraphs to make the essay more readable.

Ø  In line 90 does EMG refer to electromyography signal? This is not clearly described in the manuscript. Similarly, whether the surface electrode in Figure 3 refers to EMG electrodes, I hope the authors can clearly state this in the manuscript. If it is EMG, the specific location of the selected muscle and the sampling frequency of EMG should be described in the manuscript.

Ø  In line 100, equations 7,8 seem to be 5,6. Then it is not clear what variables VF' and Ve' refer to in equations 5,6.

Ø  Section 2.2.2 is suggested to be rewritten and I think there are a few points that are not very clear.

1. How was the C value changed? Is it determined by the experimenter's upper and lower teeth occlusion status?

2. What is the process of the experiment? For example, is the subject asked to vary the process from C=0 to C=1? Line 116 states that the subject is clenched freely, does the fact that everyone has a different rhythm introduces additional noise during the data analysis?

3. “This situation is called no clenching.” It is not clear what kind of situation is being referred to.

4. The leftmost diagram in Figure 3 would be better if the temporal and masticatory muscles were drawn.

Ø  For the participants of the experiment, basic information should be stated in the text, such as the distribution of age, gender, etc. In the process of experimental design, why was only subject E set up for an additional set of experiments?

Ø  In Figure 6, the target of C=0.0, it seems that subject E could not complete it well, and from Figure 6, it seems that subject E could not maintain the C=0 state. Is this a single case, or do most of the subjects have difficulty in completing the C=0 state?

Ø  The role of AA sum and AA ratio in the experiments defined in Equations (1), (2) is not explained in the text. And the role of the surface electrode is to control the stopping switch of the experiment as seen in Figures 3 and 5. Does the control of the actuator during the experiment depend entirely on the earable Ω (surface electrode is only used to stop the experiment)?

Ø  The results of this experiment lacked the necessary statistical analysis to verify that the experimental results were consistent with being statistically significant.

Ø  From Fig. 8, the experimental results of different subjects vary greatly, and also the NRMSE values in Fig. 9 are low, why this phenomenon occurs? Does it mean that earable Ω does not perform well in experiment 2 and is not suitable for joint stiffness control?

Ø  In the discussion section, the authors can compare their results with other literature that has used the clenching interface in recent years to highlight the innovation and effectiveness of the method proposed in this paper.

Author Response

Dear Reviewer 2.

Thank you for taking the time out of your busy schedule to read our manuscript.
We sincerely appreciate your insightful comments.
We have renumbered and reorganized the comments made by each of the two reviewers into a matrix, and have included our responses to each comment.

Round 2

Reviewer 2 Report

The authors have addressed all of my comments.

But there is one more suggestion: In Figure 9, the authors presented the averages and SDs in the graph without a detailed description of static analysis. It would be better if the authors use a table to show the detailed statistical data (averages, SDs, and P values).

Author Response

Thank you for your suggestion. Table 3 has been added in the revised manuscript according to your suggestion.

P10 Line 252, P12 Table 3.